# Optimal Control of Partially Observable Markov Decision Processes with Finite Linear Temporal Logic Constraints

**Krishna C. Kalagarla**[1]   **Dhruva Kartik**[1]   **Dongming Shen**[1]   **Rahul Jain**[1]   **Ashutosh Nayyar**[1]   **Pierluigi Nuzzo**[1]

[1]Department of Electrical and Computer Engineering, University of Southern California, Los Angeles, CA, USA

## Abstract

Autonomous agents often operate in environments where the state is partially observed. In addition to maximizing their cumulative reward, agents must execute complex tasks with rich temporal and logical structures. These tasks can be expressed using temporal logic languages like finite linear temporal logic (LTL$_f$). This paper, for the first time, provides a structured framework for designing agent policies that maximize the reward while ensuring that the probability of satisfying the temporal logic specification is sufficiently high. We reformulate the problem as a constrained partially observable Markov decision process (POMDP) and provide a novel approach that can leverage off-the-shelf unconstrained POMDP solvers for solving it. Our approach guarantees approximate optimality and constraint satisfaction with high probability. We demonstrate its effectiveness by implementing it on several models of interest.

## 1 INTRODUCTION

Markov Decision Processes (MDPs) [Puterman, 1994] can model a wide range of scenarios involving sequential decision-making in dynamically evolving environments. They are often used in settings like robotics, cyber-physical systems, and safety-critical autonomous systems. Traditional planning in MDPs involves a reward structure over the state-action space whose cumulative sum over the time-horizon is maximized to achieve a desired objective. This approach has been successful for tasks like reachability and obstacle avoidance. However, designing an appropriate reward function can at times be tricky, and an incorrect reward formulation can easily lead to unsafe and undesired behaviors. This is primarily due to the fact that instantaneous rewards in MDPs depend only on the current *system state*

and the agent's current action. When the agent's task is characterized by complex temporal objectives, the agent needs to track the *status of the task* it is performing in addition to the system state. One might be able to incorporate some of the simpler task specifications by appropriately modifying the MDP model (e.g., by adding an absorbing state that denotes obstacle collision). However, manually constructing an MDP reward function that captures substantially complicated specifications is not always possible.

To overcome this issue, increasing attention has been directed over the past decade towards leveraging temporal logic specifications [Baier and Katoen, 2008] and formal methods to formulate and solve control and planning problems in the presence of uncertainty. Several temporal logics exist that are capable of capturing a wide range of task specifications, including surveillance, reachability, safety, and sequentiality. The synthesis of MDP policies which maximize the probability of satisfaction of temporal logic specifications has also been extensively studied [Ding et al., 2011, Lahijanian et al., 2011, Aksaray et al., 2016]. However, while certain objectives are well expressed by temporal logic constraints, others are better framed as a "soft" reward maximization task. Therefore, several recent efforts [Kalagarla et al., 2021b,a, Guo and Zavlanos, 2018] have focused on reward maximization objectives for MDPs together with temporal logic constraints.

MDPs model environments where the states are fully observable and do not account for many real-life scenarios with partial state observability. These scenarios can instead be captured by Partially Observable Markov Decision Processes (POMDPs). Unfortunately, however, the aforementioned methods for synthesizing policies that satisfy temporal logic specifications in MDPs cannot be directly applied to the setting of POMDPs. In theory, any POMDP can be translated into an equivalent MDP whose state is the agent's posterior belief on the system state [Bertsekas, 1995]. However, the reachable belief space grows exponentially with the time horizon. Due to this extremely large belief space, the synthesis methods developed for MDPs become intractable

*Accepted for the 38ᵗʰ Conference on Uncertainty in Artificial Intelligence* (UAI 2022).

in the context of POMDPs.

Recently, a few approaches have been proposed to address the complexity issues that arise in POMDP planning for temporal logic specifications. The focus of these approaches is to *maximize the satisfaction* of temporal logic specifications. They include simulations over the belief space [Haesaert et al., 2018], discretization of the belief space [Norman et al., 2015], and restricting the space of policies to finite state controllers [Ahmadi et al., 2020, Sharan and Burdick, 2014, Chatterjee et al., 2015]. However, none of the above approaches addresses temporal logic and reward maximization objectives simultaneously. Lately, deep recurrent neural network based approaches [Carr et al., 2020, 2019] have also been proposed to handle POMDPs with temporal logic specifications.

In this paper, we address this problem by expanding the traditional POMDP framework to incorporate temporal logic specifications. Specifically, we aim to design policies for the agent such that the agent's reward is maximized while ensuring that the temporal logic specification is satisfied with high probability. Our focus is on processes which eventually stop, but we allow for the stopping time of the process to be random. The rewards are accumulated and the temporal logic specification must be satisfied over the duration of the process.

We focus on finite linear temporal logic ($LTL_f$) [De Giacomo and Vardi, 2013], a temporal extension of propositional logic, to express complex task specifications. $LTL_f$ is a variant of linear temporal logic (LTL) [Baier and Katoen, 2008], interpreted over finite traces. In $LTL_f$, one can start with simple atomic predicates and compose them using operators such as conjunction, negation, "until," "always," if-then, "next" (immediately), to obtain richer specifications. For example, starting with the atomic predicates "injured individual found," "seek help," and "hit obstacle," we can construct the specification "Always do not (hit obstacle) and, if (injured individual found), then immediately (seek help)." Given an $LTL_f$ specification, a deterministic finite automaton (DFA) can be constructed such that the agent's trajectory satisfies the specification if and only if it is accepted by the DFA [Zhu et al., 2017]. The internal state of this DFA essentially tracks the status of the task associated with the $LTL_f$ formula. The key idea underlying our approach is that augmenting the system state with the DFA's internal state enables us to track both the system as well as the status of our task. We can then simultaneously reason about the POMDP rewards and the temporal logic specification by formulating the planning problem as a constrained POMDP problem.

We provide a scheme which can use any off-the-shelf unconstrained POMDP solver [Kurniawati et al., 2008, Somani et al., 2013, Silver and Veness, 2010] to solve the constrained POMDP problem, thus leveraging existing results from unconstrained POMDP planning. This idea of leveraging well-studied unconstrained POMDP planners was also used to find policies maximizing temporal logic satisfaction in POMDPs [Liu et al., 2021, Bouton et al., 2020].

There are a few other approaches for solving constrained POMDPs. One such approach is to iteratively construct linear programs Poupart et al. [2015] which results in an approximate solution for the constrained POMDP problem. However, this method has been shown to suffer from scalability issues Lee et al. [2018]. A primal-dual approach based on Monte Carlo Tree Search (MCTS) Lee et al. [2018] has been used to address these scalability issues. We solve the constrained POMDP problem using a similar primal-dual method. A key difference is that, instead of using an MCTS approach, we use an approximate unconstrained POMDP solver, SARSOP Kurniawati et al. [2008]. This solver returns policies for unconstrained POMDPs along with bounds on their optimality gaps. This enables us to establish a concrete relationship between the number of iterations required and the approximation error using principles from no-regret learning. Column generation algorithms Walraven and Spaan [2018] also use a similar primal-dual approach, but with a different dual parameter update procedure. In these algorithms, convergence to optimality is shown, but the number of iterations required to get an approximate solution is not known. Our method, on the other hand, gives a precise relationship between the approximation error and the number of iterations.

To the best of our knowledge, this is the first paper on the synthesis of reward optimal POMDP policies with temporal logic constraints. Our contributions can be summarized as follows:

1. We formulate a novel problem of reward maximization in POMDPs under $LTL_f$ constraints. This formulation can incorporate several non-trivial specifications such as ordering, reactivity, etc., which cannot be well expressed by classical POMDP reward-like constraints.

2. For POMDPs that stop in finite time almost surely, we provide a structured methodology for synthesizing approximately optimal policies which maximize a cumulative reward under the constraint that the probability of satisfying a temporal logic specification stated as an $LTL_f$ formula is beyond a desired threshold.

3. We construct a constrained product POMDP expressing both the reward maximization and temporal logic objectives. We show that solving this constrained POMDP is equivalent to solving the original POMDP problem with the $LTL_f$ constraint.

4. For a large class of stopping times, we provide a planning scheme to solve the constrained POMDP. This scheme can leverage any off-the-shelf approximate solver that can solve *unconstrained* POMDPs with

stopping times. Different from current works on constrained POMDPs, we provide theoretical guarantees on the near-optimality of the returned policy by using a no-regret online learning approach.

5. Unconstrained POMDP solvers in a general stopping time setting are uncommon. We describe two specific models of stopping times for which existing POMDP solvers can be used: (i) fixed-horizon stopping and (ii) geometric stopping. Our algorithm employs a finite-horizon POMDP solver under case (i) and a *discounted* infinite-horizon POMDP solver under case (ii).

6. We apply our approach to numerically solve several models and discuss its effectiveness.

## 2 PRELIMINARIES

We denote the sets of real and natural numbers by $\mathbb{R}$ and $\mathbb{N}$, respectively. $\mathbb{R}_{\geq 0}$ is the set of non-negative reals. For a given finite set $S$, $S^*$ denotes the set of all finite sequences taken from $S$. The indicator function $\mathbb{1}_S(s)$ evaluates to 1 when $s \in S$ and 0 otherwise. For a singleton set $\{s_0\}$, we will denote $\mathbb{1}_{\{s_0\}}(s)$ with $\mathbb{1}_{s_0}(s)$ for simplicity. The probability simplex over the set $S$ is denoted by $\Delta S$. For a string $s$, $|s|$ denotes the length of the string.

### 2.1 LABELED POMDPS

**Model.** A Labeled Partially Observable Markov Decision Process (POMDP) is defined as a tuple $\mathscr{M} = (S, A, P, \varpi, O, Z, AP, L, r, T)$, where $S$ is a finite state space, $A$ is a finite action space, $P_t : S \times A \to \Delta S$ is the transition probability function at time $t$, such that $P_t(s, a; s')$ is the probability of transitioning from state $s$ to state $s'$ on taking action $a$, $\varpi \in \Delta S$ is the initial state distribution, $O$ is a finite observation space, $Z_t : S \to \Delta O$ is the observation probability function, such that $Z_t(s; o)$ is the probability of seeing observation $o$ in state $s$ at time $t$, $AP$ is a set of atomic propositions, e.g., indicating the truth value of the presence of an obstacle, goal, etc., $L : S \to 2^{AP}$ is a labeling function which indicates the set of atomic propositions which are true in each state, e.g., $L(s) = (a)$ indicates that only the atomic proposition $a$ is true in state $s$, $r_t : S \times A \to \mathbb{R}$ is a reward function, such that $r_t(s, a)$ is the reward obtained on taking action $a \in A$ in state $s \in S$. $S_t, A_t, O_t$ denote the state, action, and observation at time $t$, respectively. We say that the system is time-invariant when the reward function $r_t$ and the transition and observation probability functions $P_t$ and $Z_t$ do not depend on time $t$. The POMDP runs for a random time horizon $T$. This random time may be determined exogenously (independently) of the POMDP or it may be a stopping time with respect to the information process $\{I_t : t \geq 0\}$.

**Pure and Mixed Policies.** At any given time $t$, the information available to the agent is the collection of all the observations $O_{0:t}$ and all the past actions $A_{0:t-1}$. We denote this information with $I_t = \{O_{0:t}, A_{0:t-1}\}$. A *control law* $\pi_t$ maps the information $I_t$ to an action in the action space $A$, i.e., $A_t = \pi_t(I_t)$. The collection of control laws $\pi := (\pi_0, \pi_1, \dots)$ over the entire horizon is referred to as a *policy*. We refer to such deterministic policies as pure policies and denote the set of all pure policies with $\mathcal{P}$.

A mixed policy $\mu$ is a distribution on a finite collection of pure policies. Under a mixed policy $\mu$, the agent randomly selects a pure policy $\pi \in \mathcal{P}$ with probability $\mu(\pi)$ before the POMDP begins. The agent uses this randomly selected policy to select its actions during the course of the process. More formally, $\mu : \mathcal{P} \to [0, 1]$ is a mapping. The support of the mixture $\mu$ is defined as

$$\text{supp}(\mu) := \{\mu : \mu(\pi) \neq 0, \pi \in \mathcal{P}\}. \quad (1)$$

The set $\mathcal{M}_p$ of all mixed mappings is given by

$$\mathcal{M}_p := \left\{ \mu : |\text{supp}(\mu)| < \infty, \sum_{\pi \in \text{supp}(\mu)} \mu(\pi) = 1 \right\}. \quad (2)$$

Clearly, the set $\mathcal{M}_p$ of mixed strategies is convex.

**Assumption 1.** *The POMDP $\mathscr{M}$ is such that for every pure policy $\pi$, the expected value of the stopping time $T$ is finite, i.e.,*

$$\mathbb{E}_\pi^{\mathscr{M}}[T] < T_{\text{MAX}}^{\mathscr{M}} < \infty, \ \forall \pi. \quad (3)$$

Assumption 1 ensures that the stopping time $T$ is finite almost surely, i.e., $\mathbb{P}_\mu^{\mathscr{M}}[T < \infty] = 1$ and the total expected reward $\mathcal{R}^{\mathscr{M}}(\mu) < \infty$ for every policy $\mu$.

A *run* $\xi$ of the POMDP is the sequence of states and actions $(s_0, a_0)(s_1, a_1) \dots (s_T, a_T)$. We consider both $T$ finite as well as $T = \infty$. The total expected reward associated with a policy $\mu$ is given by

$$\mathcal{R}^{\mathscr{M}}(\mu) = \mathbb{E}_\mu^{\mathscr{M}} \left[ \sum_{t=0}^{T} r_t(S_t, A_t) \right] \quad (4)$$

$$= \sum_{\pi \in \text{supp}(\mu)} \left[ \mu(\pi) \mathbb{E}_\pi^{\mathscr{M}} \left[ \sum_{t=0}^{T} r_t(S_t, A_t) \right] \right]. \quad (5)$$

Note the $\mathcal{R}^{\mathscr{M}}(\mu)$ is a linear function in $\mu$.

### 2.2 FINITE LINEAR TEMPORAL LOGIC SPECIFICATION

We use LTL$_f$ [De Giacomo and Vardi, 2013], a temporal extension of propositional logic, to express complex

task specifications. This is a variant of linear temporal logic (LTL) [Baier and Katoen, 2008] interpreted over finite traces. Given a set $AP$ of atomic propositions, i.e., Boolean variables that have a unique truth value (true or false) for a given system state, $\text{LTL}_f$ formulae are constructed inductively as follows:

$$\varphi := \text{true} \mid a \mid \neg\varphi \mid \varphi_1 \wedge \varphi_2 \mid \mathbf{X}\varphi \mid \varphi_1\mathbf{U}\varphi_2,$$

where $a \in AP$, $\varphi$, $\varphi_1$, and $\varphi_2$ are LTL formulae, $\wedge$ and $\neg$ are the logic conjunction and negation, and $\mathbf{U}$ and $\mathbf{X}$ are the *until* and *next* temporal operators. Additional temporal operators such as *eventually* ($\mathbf{F}$) and *always* ($\mathbf{G}$) are derived as $\mathbf{F}\varphi := \text{true}\mathbf{U}\varphi$ and $\mathbf{G}\varphi := \neg\mathbf{F}\neg\varphi$. For example, $\varphi = \mathbf{F}a \wedge (\mathbf{G}\neg b)$ expresses the specification that a state where atomic proposition $a$ holds true has to be *eventually* reached by the end of the trajectory and states where atomic proposition $b$ hold true have to be *always* avoided.

$\text{LTL}_f$ formulae are interpreted over finite-length words $w = w_0 w_1 \dots w_{last} \in (2^{AP})^*$, where each letter $w_i$ is a set of atomic propositions and $last = |w| - 1$ is the index of the last letter of the word $w$. Given a finite word $w$ and $\text{LTL}_f$ formula $\varphi$, we inductively define when $\varphi$ is *true* for $w$ at step $i$, $(0 \le i < |w|)$, written $w, i \models \varphi$, as follows:

$w, i \models \text{true}$,

$w, i \models a$ iff $a \in w_i$,

$w, i \models \varphi_1 \wedge \varphi_2$ iff $w, i \models \varphi_1$ and $w, i \models \varphi_2$,

$w, i \models \neg\varphi$ iff $w, i \not\models \varphi$,

$w, i \models \mathbf{X}\varphi$ iff $i + 1 < |w|$ and $w, i + 1 \models \varphi$,

$w, i \models \varphi_1\mathbf{U}\varphi_2$ iff $\exists k$ s.t. $i \le k < |w|$ and $w, k \models \varphi_2$ and $\forall j, i \le j < k, w, j \models \varphi_1$,

$w, i \models \mathbf{G}\varphi$ iff $\forall j, i \le j < |w|, w, j \models \varphi$,

$w, i \models \mathbf{F}\varphi$ iff $\exists j, i \le j < |w|$ s.t. $w, j \models \varphi$,

where iff is shorthand for 'if and only if.' A formula $\varphi$ is *true* in $w$, denoted by $w \models \varphi$ iff $w, 0 \models \varphi$.

Given a POMDP $\mathscr{M}$ and an $\text{LTL}_f$ formula $\varphi$, a run $\xi = s_0, a_0, s_1, a_1 \dots s_T, a_T$ of the POMDP under policy $\mu$ is said to satisfy $\varphi$ if the word $w = L(s_0)L(s_1)\dots \in (2^{AP})^{T+1}$ generated by the run satisfies $\varphi$. The probability that a run of $\mathscr{M}$ satisfies $\varphi$ under policy $\mu$ is denoted by $\mathbb{P}_\mu^{\mathscr{M}}(\varphi)$.

We refer the reader to the experimental Section 5 for various examples of $\text{LTL}_f$ specifications, especially ones expressing sequentiality, which cannot be expressed by standard reward functions.

### 2.3 DETERMINISTIC FINITE AUTOMATON (DFA)

The language defined by an $\text{LTL}_f$ formula, i.e., the set of words satisfying the formula, can be captured by a Deterministic Finite Automaton (DFA) [Zhu et al., 2017].

We denote a DFA by a tuple $\mathscr{A} = (Q, \Sigma, q_0, \delta, F)$, where $Q$ is a finite set of states, $\Sigma$ is a finite alphabet, $q_0 \in Q$ is an initial state, $\delta : Q \times \Sigma \to Q$ is a transition function, and $F \subseteq Q$ is the set of accepting states.

A run $\xi_\mathscr{A}$ of $\mathscr{A}$ over a finite word $w = w_0 \dots w_n$, (with $w_i \in \Sigma$) is accepting if and only if there exists a sequence of states, $q_0 q_1 \dots q_{n+1} \in Q^{n+1}$ such that $q_{i+1} = \delta(q_i, w_i), i = 0, \dots, n$ and $q_{n+1} \in F$. A word $w \in \Sigma^*$ is accepted by $\mathcal{A}$ if and only if there exists an accepting run $\xi_\mathscr{A}$ of $\mathcal{A}$ on $w$.

Finally, we say that an $\text{LTL}_f$ formula is equivalent to a DFA $\mathscr{A}$ if and only if the language defined by the formula is the language accepted by $\mathscr{A}$. For any $\text{LTL}_f$ formula $\varphi$ over $AP$, we can construct an equivalent DFA with input alphabet $2^{AP}$ [Zhu et al., 2017].

## 3 PROBLEM FORMULATION AND SOLUTION STRATEGY

Given a labeled POMDP $\mathscr{M}$ and an $\text{LTL}_f$ specification $\varphi$, our objective is to design a policy $\mu$ that maximizes the total expected reward $\mathcal{R}^{\mathscr{M}}(\mu)$ while ensuring that the probability $\mathbb{P}_\mu^{\mathscr{M}}(\varphi)$ of satisfying the specification $\varphi$ is at least $1 - \delta$. More formally, we would like to solve the following constrained optimization problem

$$\textbf{LTL}_f\textbf{-POMDP:} \quad \max_\mu \quad \mathcal{R}^{\mathscr{M}}(\mu) \qquad \text{(P1)}$$
$$\text{s.t.} \quad \mathbb{P}_\mu^{\mathscr{M}}(\varphi) \ge 1 - \delta.$$

If (P1) is feasible, then we denote its optimal value with $\mathcal{R}^*$. If (P1) is infeasible, then $\mathcal{R}^* = -\infty$.

### 3.1 CONSTRAINED PRODUCT POMDP

Given the labeled POMDP $\mathscr{M}$ and a DFA $\mathscr{A}$ capturing the $\text{LTL}_f$ formula $\varphi$, we follow the construction by [Ding et al., 2013] for MDPs to construct a constrained product POMDP $\mathscr{M}^\times = (S^\times, A^\times, P^\times, s_0^\times, r^\times, r^f, \varpi, O, Z^\times)$ which incorporates the transitions of $\mathscr{M}$ and $\mathscr{A}$, the observations and the reward function of $\mathscr{M}$, and the acceptance set of $\mathscr{A}$.

In the constrained product POMDP $\mathscr{M}^\times$, $S^\times = (S \times Q)$ is the set of states, $A^\times = A$ is the action set, and $s_0^\times = (s_0, q_0)$ is the initial state, where $s_0$ is drawn from the distribution $\varpi$ and $q_0$ is the initial state of the DFA. For each $s, s' \in S, q, q' \in Q$, and $a \in A$, we define the transition function $P_t^\times((s, q), a; (s', q'))$ at time $t$ as

$$\begin{cases} P_t(s, a; s'), & \text{if } q' = \delta(q, L(s)), \\ 0, & \text{otherwise.} \end{cases} \qquad (6)$$

The reward functions are defined as

$$r_t^\times((s, q), a) = r_t(s, a), \ \forall s, q, a, \qquad (7)$$

$$r^f((s,q)) = \begin{cases} 1, & \text{if } q \in F \\ 0, & \text{otherwise.} \end{cases} \tag{8}$$

The observation space $O$ is the same as in the original POMDP $\mathscr{M}$. The observation probability function $Z^\times((s,q);o)$ is defined as $Z(s;o)$ for every $s \in S, q \in Q, o \in O$. We denote the state of the product POMDP $\mathscr{M}^\times$ at time $t$ with $X_t = (S_t, Q_t)$ in order to avoid confusion with the state $S_t$ of the original POMDP $\mathscr{M}$.

At any given time $t$, the information available to the agent is $I_t = \{O_{0:t}, A_{0:t-1}\}$. Control laws and policies in the product POMDP are the same as in the original POMDP $\mathscr{M}$. We define two reward functions in the product POMDP: (i) a reward $\mathcal{R}^{\mathscr{M}^\times}(\mu)$ associated with the original POMDP $\mathscr{M}$, and (ii) a reward $\mathcal{R}^f(\mu)$ associated with reaching an accepting state in the DFA $\mathscr{A}$. The reward $\mathcal{R}^{\mathscr{M}^\times}(\mu)$ is defined as

$$\mathcal{R}^{\mathscr{M}^\times}(\mu) = \mathbb{E}_\mu \left[ \sum_{t=0}^{T} r_t^\times(X_t, A_t) \right]. \tag{9}$$

The reward $\mathcal{R}^f(\mu)$ is defined as

$$\mathcal{R}^f(\mu) = \mathbb{E}_\mu \left[ r^f(X_{T+1}) \right]. \tag{10}$$

Due to Assumption 1, the stopping time $T$ is finite almost surely, and therefore, the reward $\mathcal{R}^f(\mu)$ is well-defined.

In the constrained product POMDP, we are interested in solving the following constrained optimization problem

$$\text{C-POMDP:} \quad \max_\mu \quad \mathcal{R}^{\mathscr{M}^\times}(\mu) \tag{P2}$$
$$\text{s.t.} \quad \mathcal{R}^f(\mu) \geq 1 - \delta.$$

**Theorem 1** (Equivalence of Problems (P1) and (P2))**.** *For any policy $\mu$, we have*

$$\mathcal{R}^{\mathscr{M}^\times}(\mu) = \mathcal{R}^{\mathscr{M}}(\mu) \tag{11}$$
$$\mathcal{R}^f(\mu) = \mathbb{P}_\mu^{\mathscr{M}}(\varphi). \tag{12}$$

*Therefore, a policy $\mu^*$ is an optimal solution in Problem (P1) if and only if it is an optimal solution to Problem (P2).*

*Proof.* See Appendix A. $\qquad\square$

# 4 A NO-REGRET LEARNING APPROACH FOR SOLVING THE CONSTRAINED POMDP

Problem (P2) is a POMDP policy optimization problem with constraints. Solving unconstrained optimization problems is generally easier than solving constrained optimization problems. In this section, we describe a general methodology that reduces the constrained POMDP optimization problem

(P2) to a series of unconstrained POMDP problems. These unconstrained solvers can be solved using any off-the-shelf solver. The main idea is to first transform Problem (P2) into a max-min problem using the Lagrangian function. This max-min problem can then be solved approximately using a no-regret algorithm such as the exponentiated gradient (EG) algorithm.

The Lagrangian function associated with Problem (P2) is

$$L(\mu, \lambda) = \mathcal{R}^{\mathscr{M}^\times}(\mu) + \lambda(\mathcal{R}^f(\mu) - 1 + \delta). \tag{13}$$

Let

$$l^* := \sup_\mu \inf_{\lambda \geq 0} L(\mu, \lambda). \tag{P3}$$

The constrained optimization problem in (P2) is equivalent to the sup-inf optimization problem above [Boyd and Vandenberghe, 2004]. That is, if an optimal solution $\mu^*$ exists in problem (P2), then $\mu^*$ is a maximizer in (P3), and if (P2) is infeasible, then $l^* = -\infty$. Further, the optimal value of Problem (P2) is equal to $l^*$. Consider the following variant of (P3) wherein the Lagrange multiplier $\lambda$ is bounded:

$$l_B^* := \sup_\mu \inf_{0 \leq \lambda \leq B} L(\mu, \lambda). \tag{P4}$$

**Lemma 1.** *Let $\bar{\mu}$ be an $\epsilon$-optimal strategy in sup-inf problem (P4), i.e.,*

$$l_B^* \leq \inf_{0 \leq \lambda \leq B} L(\bar{\mu}, \lambda) + \epsilon, \tag{14}$$

*for some $\epsilon > 0$. Then, we have*

$$\mathcal{R}^{\mathscr{M}^\times}(\bar{\mu}) \geq \mathcal{R}^* - \epsilon, \quad \text{and} \tag{15}$$
$$\mathcal{R}^f(\bar{\mu}) \geq 1 - \delta - \epsilon^f, \tag{16}$$

*where $\epsilon^f = \frac{R_m - \mathcal{R}^* + \epsilon}{B}$ and $R_m := \sup_\mu \mathcal{R}^{\mathscr{M}^\times}(\mu)$ is the maximum achievable reward.*

*Proof.* See Appendix B. $\qquad\square$

Lemma 1 suggests that if we can find an $\epsilon$-optimal mixed policy $\bar{\mu}$ of the sup-inf problem (P3), then the policy $\bar{\mu}$ is approximately optimal and satisfies the constraint approximately with respect to (P2), and therefore, Problem (P1) due to Theorem 1.

We use the exponentiated gradient (EG) algorithm to find an $\epsilon$-approximate policy $\bar{\mu}$ for Problem (P4). Let $f(\lambda) = \sup_\mu L(\lambda, \mu)$. A sub-gradient of the function $f(\cdot)$ at $\lambda$ is given by $(\mathcal{R}^f(\mu_\lambda) - 1 + \delta)$, where

$$\mu_\lambda = \arg\sup_\mu L(\mu, \lambda). \tag{17}$$

**Remark 1.** *For solving an unconstrained POMDP, it is sufficient to consider pure strategies, and therefore, most solvers optimize only over the space of pure strategies. Thus, the support of $\mu_\lambda$ is 1 for every $\lambda$.*

**Algorithm 1** Exponentiated Gradient Algorithm

---

Input: Constrained product POMDP $\mathcal{M}^\times$
Initialize $\lambda_1 = B/2$
**for** $k = 1, \ldots, K$ **do**
$\quad \mu_k \leftarrow \text{OPT}(\mathcal{M}^\times, \lambda_k) = \arg\sup_\mu L(\mu, \lambda_k)$
$\quad \hat{p}_k \leftarrow \text{EVAL}(\mu_k) = \mathcal{R}^f(\mu_k)$
$\quad \lambda_{k+1} = B \frac{\lambda_k e^{-\eta(\hat{p}_k - 1 + \delta)}}{B + \lambda_k(e^{-\eta(\hat{p}_k - 1 + \delta)} - 1)}$
**end for**
Output: $\bar{\mu} = \frac{\sum_{k=1}^K \mu_k}{K}$, $\bar{\lambda} = \frac{\sum_{k=1}^K \lambda_k}{K}$

---

The EG algorithm uses this sub-gradient to iteratively update $\lambda$. The value of $\lambda$ at the $k$-th iteration is denoted by $\lambda_k$ and the corresponding maximizing policy $\mu_{\lambda_k}$ is simply denoted by $\mu_k$. The EG algorithm is described in detail in Algorithm 1. Computing the sub-gradient involves two key steps: solving the unconstrained POMDP in (17) and evaluating the constraint $\mathcal{R}^f(\mu)$. The algorithm does not depend on which methods are used for solving the unconstrained POMDP and evaluating the constraint.

The following theorem states that the average policy $\bar{\mu}$ obtained from Algorithm 1 is an $\epsilon$-optimal policy for Problem (P5).

**Theorem 2.** *Under Assumption 1 and if $\eta = \sqrt{\frac{\log 2}{2KB^2}}$, the strategy $\bar{\mu}$ returned by Algorithm 1 satisfies*

$$l_B^* \leq \inf_{0 \leq \lambda \leq B} L(\bar{\mu}, \lambda) + 2B\sqrt{2\log 2 / K}. \quad (18)$$

*Therefore,*

$$\mathcal{R}^\mathcal{M}(\bar{\mu}) \geq \mathcal{R}^* - 2B\sqrt{2\log 2 / K} \quad (19)$$

$$\mathbb{P}_{\bar{\mu}}^\mathcal{M}(\varphi) \geq 1 - \delta + \frac{\mathcal{R}^* - R_m - 2B\sqrt{2\log 2 / K}}{B}. \quad (20)$$

*Proof.* The proof of this theorem is a variation of the proof of the Von Neumann theorem in Section 8.3 of [Hazan et al., 2016]. See Appendix C for details. □

In Theorem 2, we implicitly assume that Algorithm 1 has access to an exact unconstrained POMDP solver and a method for evaluating $\mathcal{R}^f(\mu)$ exactly. In practice, however, methods for solving POMDPs and evaluating policies are approximate. A similar result as in Theorem 2 can be obtained even with approximate solvers by using the arguments in Appendix D of [Kalagarla et al., 2021a].

## 4.1 FIXED STOPPING TIME

Consider the case when the horizon $T$ is a constant. With a slight abuse of notation, we denote this constant with $T$. In this case, Assumption 1 is trivially true, and therefore,

Theorem 2 holds. The Lagrangian function in this case is given by

$$L(\mu, \lambda) \quad (21)$$
$$= \mathbb{E}_\mu\left[\left(\sum_{t=0}^T r_t^\times(X_t, A_t)\right) + \lambda(r^f(X_{T+1}) - 1 + \delta)\right].$$

Clearly, for a given $\lambda$, we can maximize $L(\mu, \lambda)$ over $\mu$ using a finite-horizon POMDP solver [Walraven and Spaan, 2019]. The resulting policy $\mu_\lambda$ is a pure policy (potentially time-varying) and selects actions based on the product POMDP's posterior belief where, for an instance, $x \in S^\times$, the posterior belief $b_t \in \Delta S^\times$ at time $t$, is defined as $b_t(x) = \mathbb{P}[X_t^\times = x \mid I_t]$. The constraint $\mathcal{R}^f(\mu)$ for any policy $\mu$ can be evaluated by Monte-Carlo simulation. Therefore, with the help of a finite-horizon POMDP solver and the Monte-Carlo method for constraint evaluation, we can employ Algorithm 1 to approximately solve Problem (P1).

## 4.2 GEOMETRICALLY-DISTRIBUTED TIME HORIZON

Let $\{E_t : t = 0, 1, 2, ...\}$ be a sequence of i.i.d. Bernoulli random variables with $\mathbb{P}[E_0 = 1] = 1 - \gamma$ ($\gamma < 1$). Let the time-horizon $T$ be defined as

$$T = \min\{t : E_t = 1, t = 0, 1, \cdots\}. \quad (22)$$

This stopping time $T$ has a geometric distribution with probability mass function $(1 - \gamma)\gamma^t$. The mean of this stopping time is $\gamma/(1 - \gamma)$ for every policy, and therefore, it satisfies Assumption (1). This type of stopping time is useful in situations where the process stops when an exogenous event occurs ($E_t = 1$). The occurrence time of such exogenous events is typically modeled as a geometric (memoryless) distribution. We observe that, under this stopping model, it is possible that the process stops in just a few steps (or even one step). However, when $\gamma$ is close to 1, the probability that the process stops quickly is very small. Because of this property, this geometric stopping time can also be used to approximately model bounded horizon problems with a sufficiently large $\gamma$.

We now show that solving the unconstrained POMDP in (17) reduces to solving an equivalent discounted-reward POMDP. Discounted-reward POMDP solvers have been extensively studied and several implementations of them are readily available [Kurniawati et al., 2008, Somani et al., 2013]. Therefore, we can use any off-the-shelf discounted-reward POMDP solver for this stopping model.

Let $\mathcal{M}$ be any *time-variant* POMDP and let $\mathcal{A}$ be a DFA capturing the LTL$_f$ formula $\varphi$.

**Lemma 2.** *For a given $\lambda$, maximizing $L(\mu, \lambda)$ over $\mu$ under the geometric stopping criterion is equivalent to maximizing*

*the following discounted reward*

$$\mathbb{E}_\mu \left[ \sum_{t=0}^\infty \gamma^t \left( r_t^\times(X_t, A_t) + \frac{\lambda(1-\gamma)}{\gamma} \gamma^t r^f(X_t) \right) \right]. \quad (23)$$

*Proof.* See Appendix D. □

For a given $\lambda$, we can therefore maximize $L(\mu, \lambda)$ over $\mu$ using an infinite-horizon discounted-reward POMDP solver [Kurniawati et al., 2008]. The resulting policy $\mu_\lambda$ is a pure stationary policy and selects actions based on the product POMDP's posterior belief. The discounted-solver and a Monte-Carlo estimator can be used in Algorithm 1 to solve Problem (P1) when the stopping time is geometrically distributed.

## 5 EXPERIMENTS

We consider a collection of gridworld problems in which an agent needs to maximize its reward while satisfying an $\text{LTL}_f$ specification. In all our experiments, we use the geometric stopping (discounted) setting described in Section 4.2. Our primary reason for focusing on geometric stopping is the availability of a wide range of infinite-horizon discounted-reward solvers. The focus of our experiments is to demonstrate how our approach of constructing the product POMDP and using Algorithm 1 results in behaviors that maximize the reward and satisfy the $\text{LTL}_f$ specification. We would like to emphasize that our approach can be extended to any other stopping time model as long as it has an associated unconstrained solver and a reward estimator. The computational complexity of our approach is about $K$ (number of iterations in Algorithm 1) times the complexity of solving the unconstrained POMDP and evaluating the constraint. Therefore, the scalability of our algorithm largely depends on the scalability of the methods for solving and evaluating unconstrained POMDPs.

In all of our experiments, we use the SARSOP solver for finding an approximately optimal policy $\mu_k$ at iteration $k$ of Algorithm 1, and Monte-Carlo simulations to estimate the constraint function. Additional details on the hyper-parameters and runtime used in our experiments can be found in Appendix E. We further use the online tool $\text{LTL}_f2\text{DFA}$ [Fuggitti, 2019] based on MONA [Klarlund and Møller, 2001] to generate an equivalent DFA for an $\text{LTL}_f$ formula.

### 5.1 LOCATION UNCERTAINTY

In all the experiments in this subsection, the agent's transitions in the gridworld are stochastic. That is, if the agent decides to move in a certain direction, it moves in that direction with probability 0.95 and, with probability 0.05, it

Table 1: Reward and constraint performance of the policy $\bar{\mu}$ under various models and specifications.

| Model | Spec | $\mathcal{R}^{\mathscr{M}}(\bar{\mu})$ | $\mathcal{R}^f(\bar{\mu})$ | $1-\delta$ | $B$ |
|---|---|---|---|---|---|
| $\mathscr{M}_1$ | $\varphi_1$ | 1.72 | 0.75 | 0.75 | 5 |
| $\mathscr{M}_2$ | $\varphi_1$ | 0.95 | 0.70 | 0.70 | 8 |
| $\mathscr{M}_3$ | $\varphi_2$ | 0.83 | 0.76 | 0.75 | 5 |
| $\mathscr{M}_4$ | $\varphi_3$ | 0.80 | 0.71 | 0.70 | 6 |
| $\mathscr{M}_5$ | $\varphi_4$ | 0.83 | 0.71 | 0.70 | 6 |
| $\mathscr{M}_6$ | $\varphi_5$ | 1.01 | 0.79 | 0.80 | 10 |
| $\mathscr{M}_7$ | $\varphi_6$ | 4.28 | 0.82 | 0.80 | 25 |
| $\mathscr{M}_8$ | $\varphi_1$ | 2.73 | 0.81 | 0.85 | 20 |
| $\mathscr{M}_9$ | $\varphi_4$ | 1.68 | 0.81 | 0.75 | 10 |

moves one step with uniform probability in any direction that is not opposite to its intended direction. The agent also receives a noisy observation on where it is currently located. The observation is uniformly distributed among the locations neighboring the agent's current location. The default grid size is $4 \times 4$ and the discount factor is 0.99. The details on the reward structures can be found in Appendix E.

**Reach-Avoid Tasks.** In this problem, we are interested in reaching a goal state $a$ and always avoiding dangerous states $b$. This can be specified using $\text{LTL}_f$ as $\varphi_1 = \mathbf{F}a \wedge (\mathbf{G}\neg b)$. In this case, we consider a $4 \times 4$ grid (model $\mathscr{M}_1$ with a single obstacle $b$) and an $8 \times 8$ grid (model $\mathscr{M}_2$ with two obstacles $b$).

**Ordered Tasks.** In this problem, we are interested in reaching states $a, b$, and $c$ in a certain order. If we are interested in reaching $b$ after $a$, the corresponding specification is $\varphi_2 = \mathbf{F}(a \wedge \mathbf{F}b)$. Similarly, if we want to visit $a, b$, and $c$ in that order, the specification is $\varphi_3 = \mathbf{F}(a \wedge \mathbf{F}(b \wedge \mathbf{F}c))$. Under the specification $\mathbf{F}(a \wedge \mathbf{F}b)$, it is possible that the agent visits $b$, then $a$, and then $b$. To ensure that a strict order is maintained, we can have the specification $\varphi_4 = \neg b\mathbf{U}(a \wedge \mathbf{F}b)$. These tasks were performed on models $\mathscr{M}_3, \mathscr{M}_4$, and $\mathscr{M}_5$ (see Appendix E).

**Reactive Tasks.** In this problem, we consider a more complicated specification. There are four states of interest: $a, b, c$, and $d$. The agent must eventually reach $a$ or $b$. However, if it reaches $b$, then it must visit $c$ without visiting $d$. This can be expressed as $\varphi_5 = \mathbf{F}(a \vee b) \wedge \mathbf{G}(b \rightarrow (\neg d\mathbf{U}c))$. This task was performed on model $\mathscr{M}_6$ (see Appendix E).

Another task specification is the following: eventually reach $a$; if you visit $b$ immediately after reaching $a$, then eventually visit $c$; otherwise, visit $d$. This can be expressed as $\varphi_6 = \mathbf{F}a \wedge \mathbf{G}((a\mathbf{X}b \rightarrow \mathbf{F}c) \wedge (a\mathbf{X}\neg b \rightarrow \mathbf{F}d))$. This task was performed on model $\mathscr{M}_7$ (see Appendix E).

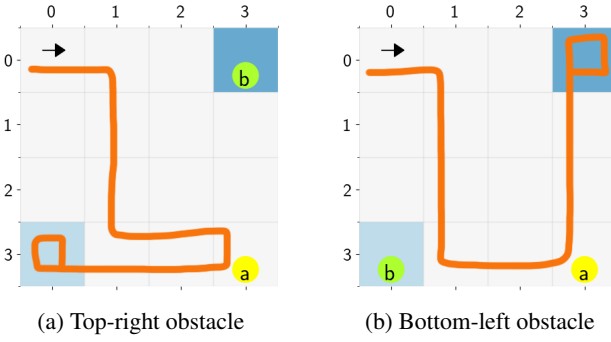

|(a) Top-right obstacle | (b) Bottom-left obstacle|

Figure 1: Trajectories in model $\mathscr{M}_8$ and specification $\varphi_1$

## 5.2 PREDICATE UNCERTAINTY

In all the experiments in this subsection, the agent's transitions in the gridworld are deterministic. That is, if the agent decides to move in a certain direction, it moves in that direction with probability 1. The uncertainty is in the location of objects that the agent may have to reach or avoid. The agent receives observations that may convey some information about an object's locations. A detailed description of the observation model is provided in Appendix E. The grid size in these models is $4 \times 4$ and the discount factor is 0.99.

**Reach-Avoid Tasks.** The reach avoid specification ($\varphi_1$) is the same as earlier. However, the agent does not know which location to avoid. The agent must therefore gather enough information to assess where the undesirable state is and act accordingly. This task was performed on model $\mathscr{M}_8$ (see Appendix E).

**Ordered Tasks.** The agent needs to visit state $a$ and $b$ strictly in that order. Therefore, the specification is $\varphi_4$. However, the agent does not know where $b$ is located. Once again, it must gather enough information and then traverse the grid accordingly. This task was performed on model $\mathscr{M}_9$ (see Appendix E).

For each model discussed above, we use Algorithm 1 to generate a mixed policy $\bar{\mu}$. The corresponding reward $\mathcal{R}^{\mathscr{M}}(\bar{\mu})$ and the constraint $\mathcal{R}^f(\bar{\mu})$ (which is the same as the satisfaction probability $\mathbb{P}_{\bar{\mu}}^{\mathscr{M}}(\varphi)$) are shown in Table 1. The reward and the constrained have been estimated by running 200 Monte-Carlo simulations. We observe that the probability of satisfying the constraint generally exceeds the required threshold. Occasionally, the constraint is violated, albeit only by a small margin. This is consistent with our result in Theorem 2. Since we cannot exactly compute the optimal feasible reward $\mathcal{R}^*$, it is difficult to assess how close our policy is to optimality. Nonetheless, we observe that the agent behaves in a manner that achieves high reward in all of these models. A more detailed discussion on this can be found in Appendix E.

## 5.3 DISCUSSION

In this section, we discuss the interplay between reward maximization, constraint satisfaction, and partial observability for executing the reach-avoid task in model $\mathscr{M}_8$. The state in this model comprises of two parts: (i) the agent's location and (ii) the object $b$'s location. The object can only be in the bottom-left corner or the top-right corner (see Figure 1). The agent receives high reward when it remains in the top-right corner, moderate reward in the bottom-left corner, and no reward everywhere else. Further, the agent does not know the obstacle's location a priori. If the agent gets close to the obstacle, it can detect the obstacle with some probability. The agent's detection capability is better when it is in the bottom-left region than when it is in the top-right region (see Appendix E).

In order to balance the reward, constraint satisfaction and information acquisition, our agent acts as follows. It first heads towards the location $a$ (since it has to eventually visit it) via the bottom-left region without hitting the corner. Since the agent's detection capability is higher in the bottom-left region, it acquires information on where the object is located. After reaching $a$, it goes to the top-right corner if the object is *not* located there and bottom-left corner otherwise. Some typical trajectories of the agent are shown in Figure 1.

Plot 2 depicts the performance of various policies $\mu_k$ generated while executing Algorithm 1. We can observe that, in the vast majority of iterations, the constraint is being satisfied. The Lagrange multiplier $\lambda_k$ decreases as long as the constraint is being satisfied. The Lagrange multiplier eventually becomes too small and the constraint is violated. This is when we observe a spike in the reward (see Figure 2). These spikes add to the average reward. Since the constraint violation is substantial, the Lagrange multiplier increases. We note that this iterative process ensures that constraint violation occurs rarely. Since we randomly pick a policy with uniform distribution, the average error probability is still close to the threshold (see Table 1).

## 6 CONCLUSIONS

In this paper, we provided a methodology for designing policies that maximize the total expected reward while ensuring that the probability of satisfying a linear temporal logic (LTL$_f$) specification is sufficiently high. By augmenting the system state with the state of the DFA associated with the LTL$_f$ specification, we constructed a constrained product POMDP. Solving this constrained product POMDP is equivalent to solving the original problem. We provided an alternative constrained POMDP solver based on the exponentiated gradient (EG) algorithm and derived approximation bounds for it. We identified two types of stopping time (fixed and geometric) for which we have readily available unconstrained POMDP solvers which can be used by

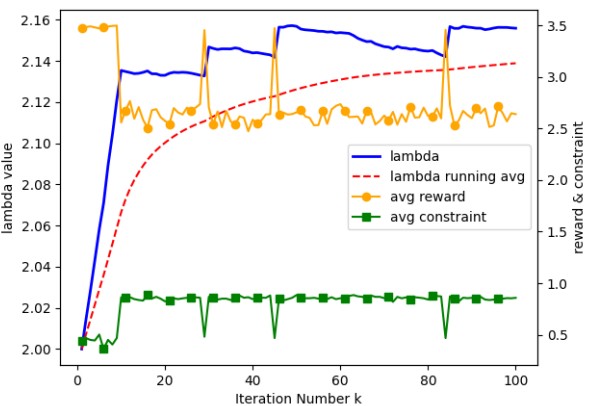

Figure 2: This plot depicts how the Lagrange multiplier $\lambda_k$, the reward $\mathcal{R}^{\mathcal{M}}(\mu_k)$ and the probability of satisfaction $\mathcal{R}^f(\mu_k)$ evolve with $k$ in Algorithm 1 under model $\mathcal{M}_8$ with the reach-avoid specification $\varphi_1$.

our constrained POMDP solver. For geometric stopping time models, we computed near optimal policies that satisfy the $\text{LTL}_f$ specification with sufficiently high probability. We observed in our experiments that our approach results in policies that effectively balance information acquisition (exploration), reward maximization (exploitation), and satisfaction of the specification, which is very difficult to achieve using classical POMDPs.

## Acknowledgements

This research was supported in part by the National Science Foundation under Awards 1839842, 1846524, and 2139982, the Office of Naval Research under Award N00014-20-1-2258, and the Defense Advanced Research Projects Agency under Award HR00112010003.

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
