# OpenReview forum: "Optimal Control of Partially Observable Markov Decision Processes with Finite Linear Temporal Logic Constraints"
_auai.org/UAI/2022/Conference — UAI 2022 Poster_

### Official Review · Reviewer_JSHC · 2022-04-10

**Q2(1) Originality/Novelty:** 3
**Q2(2) Significance/Impact:** 3
**Q2(3) Correctness/Technical Quality:** 3
**Q2(6) Clarity Of Writing:** 4
**Q6 Overall Score:** 7
**Q8 Confidence In Your Score:** 3

**Q1 Summary And Contributions:**

The paper addresses the problem of combining LTL constraints while preserving optimal reward of policies for POMDPs. A novel theoretical formulation of the problem is proposed, considering a product POMDP with constraints inherited by a deterministic finite automaton extracted from LTL specifications over finite traces. The authors propose an efficient algorithm for solving the constrained product POMDP iteratively, using state-of-the-art general-purpose unconstrained POMDP solvers.

**Q2 Assessment Of The Paper:**

More detailed information regarding each of these aspects is given below:

**Q2(4) Quality Of Experiments (Optional):**

3: Good: The experimental evaluation is adequate, and the results convincingly support the main claims.

**Q2(5) Reproducibility:**

3: Good: Key resources (e.g., proofs, code, data) are available and key details (e.g., proofs, experimental setup) are sufficiently well-described for competent researchers to confidently reproduce the main results.

**Q3 Main Strengths:**

The authors provide a sound theoretical formulation of LTL-constrained POMDP using the theory of deterministic finite automata. This allows to define a very useful practical strategy for solving the original problem by iterative solving unconstrained POMDP, which can be done by using any off-the-shelf available solver. The state of the art review is comprehensive and the writing is clear and rigorous. Experiments are meaningful, analyzing the useful scenario of geometrically-distributed time horizon and considering different LTL specifications, with both location and predicate uncertainties.

**Q4 Main Weakness:**

I am not fully convinced by the choice of delta, i.e. the bound for constraint satisfaction, in the empirical evaluation. In my opinion, this is not adequately discussed by the authors, while I think it is a relevant, though probably domain-specific, parameter. Moreover, it would be interesting to see whether and how this parameter affects the time for finding an optimal solution with the proposed iterative algorithm. In fact, in the Appendix the authors mention the runtime information related to the complexity of the domain, hence the size of finite automaton and computational burden for unconstrained POMDP solver.


**Q5 Detailed Comments To The Authors:**

I suggest the authors to clarify the choice of delta parameter in the experiments, and possible related consequences on the runtime (or even generic considerations about the relation between runtime and rate of convergence of the iterative algorithm, i.e. K value in Algorithm 1).
I report few typos that should be fixed:
- page 2, first column, middle: "until, " -> "until", ... and so on for other occurrences
- page 3, equation (3): please use a different notation for delta (DFA transition function or constraint satisfaction threshold?)
- page 5, right before Theorem 2: remove "satisfies"; Reference should be to Problem P4, not P5
- page 6, 2nd column, top: double "the"
- page 7, right below Table 1: "Algorithn" -> "Algorithm"

**Q7 Justification For Your Score:**

The authors provide both a sound theoretical formulation and a useful practical implementation for the relevant problem of optimal constrained POMDP solution. The writing is sound and rigorous.

**Q9 Complying With Reviewing Instructions:**

1: Yes.

---

### Official Review · Reviewer_b14h · 2022-04-10

**Q2(1) Originality/Novelty:** 3
**Q2(2) Significance/Impact:** 2
**Q2(3) Correctness/Technical Quality:** 2
**Q2(6) Clarity Of Writing:** 2
**Q6 Overall Score:** 4
**Q8 Confidence In Your Score:** 3

**Q1 Summary And Contributions:**

This paper describes an approach to policy optimization for POMDPs that maximize cumulative reward under the constraint that the probability of satisfying a temporal logic specification stated as an LTLf formula is beyond a desired threshold.


**Q2 Assessment Of The Paper:**

More detailed information regarding each of these aspects is given below:

**Q2(4) Quality Of Experiments (Optional):**

3: Good: The experimental evaluation is adequate, and the results convincingly support the main claims.

**Q2(5) Reproducibility:**

3: Good: Key resources (e.g., proofs, code, data) are available and key details (e.g., proofs, experimental setup) are sufficiently well-described for competent researchers to confidently reproduce the main results.

**Q3 Main Strengths:**

Two contributions:

1) Shows (for the first time) how to convert a reward-maximization POMDP with LTL specifications into a constrained (product) POMDP.

2) Describes an approach to solving a constrained POMDP by converting in into a sequence of unconstrained POMDPs, which can be solved by an off-the-shelf solver.


**Q4 Main Weakness:**

After showing how to convert a POMDP with LTL constraints into a constrained POMDP,  the paper proposes an approach to solving the constrained POMDP without any reference to, or discussion of, the large existing body of work on solving constrained POMDPs in the AI community — which makes it difficult to assess the contribution.

**Q5 Detailed Comments To The Authors:**

Section 4 describes an approach to solving a constrained POMDP by reducing it to a series of unconstrained POMDPs that can each be solved by an approximate solver.  There is considerable related work on solving constrained POMDPs, and none of it is referenced or discussed in this submission. This related work is not based on LTL constraints, but it is not clear why that should matter, since the authors show in this paper how to convert a POMDP with LTL constraints into a constrained POMDP. As one example, consider the following paper:

Walraven and Spaan (2018), “Column generation algorithms for constrained POMDPs.” JAIR vol. 62.

It also describes an approach to solving a constrained POMDP by reducing it to a series of unconstrained POMDPs that can each be solved by an approximate solver — very similar to the approach proposed in this submission. Without a discussion of related work on constrained POMDPs, it is difficult for a reader to assess the contribution of this submission and put it in perspective. I wonder if the failure to reference related work is related to the fact that the references considered in this paper are heavily slanted towards prior work in the formal methods and verification community, with apparently less awareness of the AI literature.

The paper is difficult to read, it relies a lot on references to supplementary appendices, and I wonder if it tries to include too much. The most extreme case is the last paragraph before section 4.1, which refers to a proof in an *appendix* of an *unpublished* paper to show that approximate POMDP solvers can be used to provide exact guarantees. Since the validity of the approach depend on it, it deserves more explanation! It may be another example of not being able to fit everything in one paper.

The complicated math formulas in Lemma 1 and Theorem 2 would benefit from some kind of intuitive explanation! I see that they are bounds, but otherwise, they provide little insight.

The authors write: “In all of our experiments, we use the SARSOP solver for finding the optimal policy” (first sentence on page 7).  I am confused by the claim that a point-based solver like SARSOP can find an optimal policy. Bullet 1 on page 2 also claims: “we provide a structured methodology for synthesizing optimal policies.” Again, I don’t understand the claim to find optimal policies for infinite-horizon POMDPs.

The authors claim that their approach applies to “POMDPs that stop in finite time almost surely,” and then show it applies to problems with a geometrically distributed horizon. Aren’t there POMDPs that terminate in finite time almost surely, but don’t have a geometrically distributed horizon that can be reduced to a discounted POMDP? I’m thinking in particular of a POMDP where the goal is to reach a particular state with probability 1 with minimum cost, and the structure of the problem ensures this is possible.

**Q7 Justification For Your Score:**

The topic of integrating LTL constraints in POMDP solvers is important and promising, and the authors claim to be the first to do so in a way that includes reward maximization. The proposed approach transforms the problem into an equivalent constrained POMDP, which it considers how to solve, but it does so without any reference to a large body of related work on solving constrained POMDPs, which makes its contribution difficult to assess.


**Q9 Complying With Reviewing Instructions:**

1: Yes.

---

### Official Review · Reviewer_v2y1 · 2022-04-12

**Q2(1) Originality/Novelty:** 3
**Q2(2) Significance/Impact:** 3
**Q2(3) Correctness/Technical Quality:** 3
**Q2(6) Clarity Of Writing:** 4
**Q6 Overall Score:** 8
**Q8 Confidence In Your Score:** 1

**Q1 Summary And Contributions:**

This paper contributes to an increasing volume of papers adopting LTLf in
various contexts ---which is good! In this case, the authors suggest a method
for extending POMDPs with LTLf for verifying the satisfaction of linear temporal
logic expressions with large probability.


**Q2 Assessment Of The Paper:**

More detailed information regarding each of these aspects is given below:

**Q2(4) Quality Of Experiments (Optional):**

3: Good: The experimental evaluation is adequate, and the results convincingly support the main claims.

**Q2(5) Reproducibility:**

3: Good: Key resources (e.g., proofs, code, data) are available and key details (e.g., proofs, experimental setup) are sufficiently well-described for competent researchers to confidently reproduce the main results.

**Q3 Main Strengths:**

The approach starts as most papers, i.e., with the cross-product of the DFA and
the subject of the LTLf, in this case, a POMDP and thus, even if as told the
conference organizers, this paper falls very far from my field of expertise,
truth is that the methods followed look sound to me ---with the exception of
those stages where unconstrained POMDPs are solved.

Two weak comments follow with the hope they are useful to the reviewers and also
the other PC members:

There is a typo in page 5: "Algorithm 1 *satisfies is* an \epsilon-optimal policy" -

Secondly, the experimental section looks rather weak to me, even if a foreigner.
The examples chosen are very rich and certainly convincing, and the results look
interesting, but al most no discussion is provided.


**Q4 Main Weakness:**

The approach starts as most papers, i.e., with the cross-product of the DFA and
the subject of the LTLf, in this case, a POMDP and thus, even if as told the
conference organizers, this paper falls very far from my field of expertise,
truth is that the methods followed look sound to me ---with the exception of
those stages where unconstrained POMDPs are solved.

Two weak comments follow with the hope they are useful to the reviewers and also
the other PC members:

There is a typo in page 5: "Algorithm 1 *satisfies is* an \epsilon-optimal policy" -

Secondly, the experimental section looks rather weak to me, even if a foreigner.
The examples chosen are very rich and certainly convincing, and the results look
interesting, but al most no discussion is provided.


**Q5 Detailed Comments To The Authors:**

The approach starts as most papers, i.e., with the cross-product of the DFA and
the subject of the LTLf, in this case, a POMDP and thus, even if as told the
conference organizers, this paper falls very far from my field of expertise,
truth is that the methods followed look sound to me ---with the exception of
those stages where unconstrained POMDPs are solved.

Two weak comments follow with the hope they are useful to the reviewers and also
the other PC members:

There is a typo in page 5: "Algorithm 1 *satisfies is* an \epsilon-optimal policy" -

Secondly, the experimental section looks rather weak to me, even if a foreigner.
The examples chosen are very rich and certainly convincing, and the results look
interesting, but al most no discussion is provided.


**Q7 Justification For Your Score:**

Just guessing, as I repeatedly reported this paper falls very far from my field of expertise

**Q9 Complying With Reviewing Instructions:**

1: Yes.

---

### Official Review · Reviewer_ZxcQ · 2022-04-17

**Q2(1) Originality/Novelty:** 2
**Q2(2) Significance/Impact:** 2
**Q2(3) Correctness/Technical Quality:** 3
**Q2(6) Clarity Of Writing:** 3
**Q6 Overall Score:** 5
**Q8 Confidence In Your Score:** 3

**Q1 Summary And Contributions:**

Summary: This paper provides a framework for solving an POMDP that maximize the cumulative reward while satisfying the temporal logic specification with high probability.

**Q2 Assessment Of The Paper:**

More detailed information regarding each of these aspects is given below:

**Q2(4) Quality Of Experiments (Optional):**

2: Fair: The experimental evaluation is weak: important baselines are missing, or the results do not adequately support the main claims.

**Q2(5) Reproducibility:**

2: Fair: Key resources (e.g., proofs, code, data) are unavailable but key details (e.g., proof sketches, experimental setup) are sufficiently well-described for an expert to confidently reproduce the main results.

**Q3 Main Strengths:**

1.This paper claims that it is the first work that solves POMDP and satisfies the pre-specified temporal logic rules with high probability.
2.The stopping time of the process can be either fixed (the horizon T is a constant), or random, such as following geometric distribution in this paper.
3.The paper is well-organized. The  paper provides clear a statement about the Labeled Partially Observable Markov Decision Process (POMDP), linear temporal logic and deterministic finite automation.
4.The paper describes a general methodology that reduces the constrained POMDP optimization problem to a series of unconstrained POMDP problems and provides theoretical proof.
5.The experiment part demonstrates good performance of this framework through 5 different tasks in gridworld environment. The results indicate high cumulative reward earned and high probability of satisfying rules.

**Q4 Main Weakness:**

1. Although this paper is the first work that solves POMDP and satisfies the pre-specified temporal logic rules with high probability, I believe  that all its building blocks: 1)  how to solve a POMDP/MDP by satisfying the temporal logic rules; 2) how to solve a constrained RL problem are all well-studied. This paper just pieces them together and the contribution seems to be incremental.

2.In the experiment, the default gird size is 4 by 4, which maybe too small. The paper only focuses on the gridworld environment, while it does not provide other experimental environments. In addition, the paper does not provide baselines or benchmark to compare with the proposed method, so we don’t know how well this method behaves.



**Q5 Detailed Comments To The Authors:**

1. Could the authors further clarify the novelty and technical contribution of this paper?
2. Solving POMDPs alone is a challenging problem. Is this the reason why the experiments only considered toy examples?

**Q7 Justification For Your Score:**

Overall the paper is clearly written. The problem considered is important and interesting.
The paper needs to further clarify its technical contribution compared to existing works.

**Q9 Complying With Reviewing Instructions:**

1: Yes.

---

### Decision · Program_Chairs · 2022-05-15

**Decision:**

Accept (Poster)

**Comment:**

Meta Review: This paper makes a valuable contribution, however, you need to reference related work on solving constrained POMDPs, and discuss the relationship of your work to this body of work.